# Radiomics in Soft Tissue Sarcoma: Toward Precision Imaging in Oncology

**DOI:** 10.3390/cancers17223661

**Published:** 2025-11-14

**Authors:** Anuj Shah, Francesco Alessandrino, Emanuela Palmerini, Domenika Ortiz Requena, Brooke Crawford, Ty K. Subhawong

**Affiliations:** 1Department of Radiology, University of Miami Miller School of Medicine, Miami, FL 33136, USA; 2Sylvester Comprehensive Cancer Center, Miami, FL 33136, USA; 3Department of Medicine, University of Miami Miller School of Medicine, Miami, FL 33136, USA; 4Department of Pathology, University of Miami Miller School of Medicine, Miami, FL 33136, USA; 5Department of Orthopaedics, University of Miami Miller School of Medicine, Miami, FL 33136, USA

**Keywords:** radiology, radiomics, MRI, soft tissue sarcoma, neoplasm, treatment response, biomarker

## Abstract

Radiomics allows for the extraction of quantitative information from imaging studies and is rapidly changing the field of oncologic imaging to allow for improved diagnosis and management of a variety of tumors. This review examines the application of radiomics to soft tissue sarcomas, describing how radiomics allows for improvements in initial diagnosis, grading, and prognostication of soft tissue sarcomas. Radiomics has further allowed for data to guide treatment decisions and for the development of new analytic methods to assess treatment response of these tumors. Though challenges exist in the reproducibility of studies, model development for many rare soft tissue sarcoma subtypes, and the limited integration of radiomic workflows into PACS, radiomics represents a promising avenue in the imaging of soft tissue sarcomas.

## 1. Introduction

Soft tissue sarcomas are a diverse and rare group of malignancies of mesenchymal tissues, and diagnostic imaging can play a significant role in early detection and treatment of these tumors. These tumors are often difficult to diagnose and treat due to the diversity of their imaging appearance, anatomic origin, and aggressiveness.

Imaging of a potential sarcoma, often presenting as a palpable soft tissue mass, begins with an ultrasound to evaluate the internal characteristics of the mass and determine its size, depth, and margins, as well as radiography to examine for mineralization and erosion of surrounding bone [1]. The majority of these suspicious masses, particularly those with deep invasion, large tumors, and those incompletely imaged on ultrasound, should be imaged with contrast-enhanced MRI to detail the tumor matrix and borders [2]. Additional information can further characterize the mass and point to the diagnosis of a particular type of soft tissue sarcoma, including the presence of internal fat indicative of adipocytic tumors or the triple sign seen in hemorrhagic tumors like synovial sarcoma [3,4]. Challenges in diagnosis on MRI include the heterogenous phenotypes and significant imaging overlap between benign and locally aggressive tumors (e.g., nodular fasciitis and desmoid fibromatosis) and frankly malignant sarcomas [5]. While higher grade sarcomas tend to be larger, harbor areas of central necrosis, and display peritumoral edema, many of these imaging features remain too nonspecific to reliably distinguish between malignant and benign lesions, and ultimately biopsy is required for tissue diagnosis or imaging/clinical follow-up must confirm stability [6]. CT imaging can aid in preoperative staging, and has been shown in some trials to be non-inferior to MRI in staging soft tissue sarcomas of the extremities [7,8].

Beyond conventional CT and MRI, several advanced imaging modalities can improve the noninvasive characterization of soft tissue tumors, albeit with their own drawbacks. Diffusion-weighted imaging (DWI) and apparent diffusion coefficient (ADC) mapping can provide insight into tumor cellularity and distinguish malignant from benign lesions but are limited by value overlap and susceptibility to motion and distortion artifacts [9]. Dynamic contrast-enhanced (DCE) MRI and MR spectroscopy can offer additional information about tumor perfusion and metabolic profile, but are not yet routinely used in clinical practice due to technical complexity and limited spatial resolution [9,10]. Similarly, PET-CT can assess metabolic activity and guide biopsy or staging, but its utility is limited by having lower anatomical detail compared to an MRI and a risk toward patients of repeated exposure to ionizing radiation. Additionally, it is not typically used as standard of care in soft tissue sarcomas as there are non-avid subtypes [11]. While each of these modalities contributes valuable physiologic or metabolic information, they are only complementary to high-resolution anatomic imaging and tissue sampling/biopsy. These limitations have prompted growing interest in radiomics and machine learning approaches to extract more nuanced, quantitative data from routine imaging studies to improve diagnostic accuracy and risk stratification in soft tissue sarcoma.

Traditional semantic descriptors of tumor morphology and heterogeneity, such as size, margin definition, and signal intensity (in MRI) or pixel intensity qualified as Hounsfield units (in CT), offer only a limited representation of the underlying tumor biology. While dispersion metrics like standard deviation can provide a global measure of signal variability, they fail to account for the spatial arrangement of pixel intensities within the tumor. This means that tumors with vastly different internal architectures may appear similar when assessed using only global statistics (Figure 1). Such approaches overlook crucial information about the spatial distribution of heterogeneity, such as the presence of necrotic cores, peripheral enhancement, or compartmentalization patterns that may correlate with tumor grade, aggressiveness, or treatment response. This limitation may be addressed by radiomic texture analysis, which can capture both the magnitude and spatial organization of signal variation in a more reproducible and quantitative manner. While radiologists may be limited to recognizing semantic features, radiomics allows for extraction of agnostic features describing lesion heterogeneity [12]. These agnostic features include first-order intensity statistics, second-order texture measures (such as co-occurrence and run-length matrices), and higher-order features that can describe heterogeneity across multiple different scales [12]. This vastly expands the number of descriptive variables that can be used but comes with its own challenges, specifically related to reproducibility, interpretability, and risk of overfitting [12].

Radiomics, the subfield of medical imaging that extracts a broad range of quantitative features from imaging studies using data characterization algorithms, can play a pivotal role in oncologic imaging due to its ability to uncover patterns in imaging features that may not be apparent to the human eye [13,14]. Radiomic data can be gathered from several imaging modalities, including CT, MRI, and PET, and can include features such as intensity, texture, and heterogeneity [13]. The first step in this process involves preprocessing of the images, followed by segmentation of the original exam, or highlighting a region of interest such as a tumor, from which features are to be extracted (Figure 2) [13,15]. Although segmentation was initially a manual process, parts of it can now be automated. Features are then extracted and a model can be constructed for use in classifying a particular tumor or predicting clinical outcomes. Though the utility of radiomics in general oncology appears promising, several barriers to its implementation and effectiveness exist, including the need for normalization of images and imaging protocols for standard radiomics techniques to apply, resampling of the resolution of images for analysis, the need for automation of the otherwise tedious segmentation process, and identification of robust features among hundreds of potentials that retain significance across multiple studies [12]. Furthermore, small variations in acquisition, reconstruction, or segmentation can create differences that do not truly reflect biologic differences, and these problems are amplified in rare tumors like sarcomas, where limited sample sizes restrict model validation [12]. Overcoming these barriers will require standardized pipelines, robust feature selection, and above all, multi-institutional collaboration and data sharing to ensure that radiomic biomarkers achieve reproducibility and clinical relevance. Machine learning and AI offer a potential avenue to automate and accelerate many of the components of the radiomics workflow.

When applied to soft tissue sarcomas, radiomics offers an opportunity to incorporate objective, reproducible imaging biomarkers into clinical workflows. These biomarkers could aid in distinguishing benign from malignant masses, subtyping sarcomas, predicting metastatic potential, and assessing treatment response, especially in the context of neoadjuvant therapy. This review aims to explore the current state of radiomics in the diagnosis, prognostication, treatment planning, and assessment of treatment effectiveness in soft tissue sarcomas.

## 2. Radiomics in Initial Diagnosis of Soft Tissue Sarcomas

One of the first key clinical questions radiomics must address for a particular tumor type is whether a particular malignant lesion can be distinguished from benign tumors or other cancer types. This problem is especially relevant in situations where access to the lesion for biopsy is particularly difficult, or in heterogeneous tumors prone to tissue sampling errors. While some imaging features predict diagnosis and prognosis for specific soft tissue sarcoma subtypes, including “tail sign” for myxofibrosarcoma and “triple sign” (solid cellular elements, hemorrhage or necrosis, and fibrotic regions) for synovial sarcoma, the majority of soft tissue sarcomas are unable to be labeled as a specific subtype based on imaging findings alone [16]. A recent meta-analysis of 10 studies (885 pooled subjects) evaluating the use of MRI radiomics in soft tissue sarcoma diagnosis (benign vs. malignant) yielded a sensitivity of 84%, specificity of 83%, and AUC of 0.93, suggesting a role for radiomics in selecting which tumors are referred for biopsy [17].

One of the most interesting prospects for radiomics-informed specific tumor diagnosis lies in distinguishing uterine leiomyosarcoma and leiomyoma (particularly atypical or degenerated leiomyomas), with important implications for additional work-up and management. Needle biopsy is often contraindicated in suspected uterine leiomyosarcoma as it risks tumor seeding in the peritoneum. Even without radiomics, meta-analyses show MRI can offer some useful metrics in differentiating the two tumor types. For example, leiomyosarcomas have lower ADC values than leiomyomas, more heterogenous contrast enhancement, and increased signal intensity on T1-weighted images owing to internal hemorrhage [18]. Differing data exists on the utility of radiomics in solving this particular clinical problem, but a large number of studies established unique radiomic features separating leiomyosarcomas and leiomyomas—for example, Lakhman et al. found that leiomyosarcoma tends to have greater textural heterogeneity, higher contrast, lower energy, and lower kurtosis on MRI-based texture metrics compared to leiomyomas [19]. Several groups found that radiomics models were comparable to experienced radiologists in distinguishing the diagnoses of uterine sarcoma versus leiomyoma, although many of these findings were preliminary and derived from limited single-center cohorts [20,21]. For example, in a study by Xie et al., MRI features (including ADC-maps) plus a radiomic model were used in a cohort of 78 patients (29 sarcomas, 49 leiomyomas) [21]. They found that older age, ill-defined margins, and disrupted endometrial cavity were significant clinical-MRI predictors of leiomyosarcoma and that radiomic texture features had a diagnostic performance comparable to expert radiologists [21]. Nagakawa et al. also found that radiomic data from multiparametric MRI was superior to PET and SUV values and comparable to radiologists in ability to distinguish leiomyosarcoma from leiomyoma [20]. Similarly, a model by Roller et al. found marginal improvement in predicting the difference between the two tumor types with an MRI-based model combining radiomics, conventional imaging features, and clinical data, as compared to a model utilizing MRI and clinical data only (AUC 0.989 vs. 0.956) [22]. Alternatively, a CT-based study incorporated contrast-enhanced CT and machine learning techniques and achieved an AUC of 0.78–0.97 with sensitivity up to 100% and specificity up to 93%, which outperformed radiologists alone (AUC 0.73–0.75) [23]. Another similar model improved classification for T2 hyperintense uterine mesenchymal tumors, with the combined model (AUC 0.91) outperforming radiologists (AUC 0.78–0.90), clinical data alone (AUC 0.79), and radiomics alone (AUC 0.76) [24]. These studies as a whole suggest that combining clinical data, individual expertise from experienced radiologists, and new radiomics data from MRI or CECT may improve the sensitivity and accuracy of the diagnosis of suspected uterine leiomyosarcoma on imaging. More recent studies have even demonstrated the ability of radiomics-based CT models to differentiate retroperitoneal leiomyosarcoma from other tumor types such as retroperitoneal liposarcoma [25].

The benefits of radiomic analysis seen in the case of uterine leiomyosarcoma extend to a diverse set of other soft tissue sarcoma subtypes. For example, radiomics has been able to differentiate between phenotypes of soft tissue tumors, including desmoid tumors and leiomyosarcoma, with these analyses even allowing for variations in MRI protocols [26]. For example, Timbergen et al. demonstrated the use of a radiomics model to distinguish desmoid-type fibromatosis from soft tissue sarcomas across 203 cases (AUC 0.88), comparable to two radiologists (AUC 0.8, 0.88). These principles have also been applied to MRI radiomics-based differentiation of benign versus malignant peripheral nerve sheath tumors (AUC 0.94), with several intermediate radiomics features even correlating to premalignant atypical peripheral nerve sheath tumors that lie in between the benign and malignant tumors in terms of differentiation [27]. In addition, models have been developed to differentiate nodular fasciitis versus soft tissue sarcomas, such as radiomics nomograms developed by Wang et al. to draw information from MRI data [28]. Benhabib et al. demonstrated the ability of MRI radiomics to distinguish between benign myxomas and malignant myxoid sarcomas in a cohort of 523 cases (AUC 0.92) [29].

A potential avenue for radiomics to be integrated into standard MRI imaging workflows for soft tissue sarcomas is for a radiomic risk score to be generated from initial imaging studies. For example, when an MRI-based radiomic risk score was derived from images of tumors for 176 patients and combined with age, Li et al. found improved ability to differentiate benign and malignant soft tissue tumors (AUC 0.84) [5].

This potential to improve risk stratification for malignancy is a crucial way that radiomics will help guide management decisions in cases where biopsies are non-diagnostic or when choosing surveillance strategies for indeterminate tumors.

## 3. Grading, Prognostication, and Outcomes

Histologic grade is the most important prognostic marker of soft tissue sarcomas, but an initial biopsy can often underestimate the final grade of a soft tissue sarcoma due to tumor heterogeneity and undersampling [30,31,32,33]. This makes incorporating radiomics a critical component to a more comprehensive understanding of tumor biology. Radiomics allows for both a global tumor assessment and recognition of particular tumor regions that have the potential for more aggressive behavior.

Early studies were able to utilize MRI-based radiomics to predict high vs. low grade soft tissue sarcomas with an AUC of 0.92, accuracy of 91.4%, sensitivity of 88.2%, and specificity of 94.4% [34]. Further studies have since confirmed the ability of radiomics features from MRI to predict malignancy grade by examining tumor heterogeneity on T2 and contrast-enhanced sequences, as well as peritumoral edema [35]. Groups have also been able to develop MRI-based nomograms based on intratumoral habitat features and peritumoral edema that are able to outperform the use of radiomics alone and clinical features alone (AUC 0.868 vs. 0.856) [36]. Most importantly, multicenter studies have demonstrated the ability of these models to predict histopathologic grade of soft tissue sarcomas based on intratumoral and peritumoral radiomics features, demonstrating the ability of some models to be applied across multiple different settings [37]. All in all, meta-analyses have found that MRI radiomics have a high accuracy (sensitivity 84%, specificity 73%, AUC 0.91) for distinguishing low- from high-grade soft tissue sarcomas, with machine learning techniques demonstrating the ability to surpass individual feature analysis in their predictive performance [17,38]. However, it should be noted that some of these pooled results are confounded by inconsistant choices for grade dichotomization, with some studies including FNCLCC grade 2 tumors with low-grade sarcomas, and others considering grade 2 as high-grade sarcomas [34].

When imaging a suspected soft tissue sarcoma, in addition to grade, several other elements are considered key prognostic factors. These include size of the mass, peritumoral enhancement, signs of necrosis, tumor depth, and poor definition of borders [6,16]. Schmitz et al. utilized radiomic features to identify characteristics of low- versus high-proliferative soft tissue sarcomas, and found that heterogeneity and ill-defined margins were more commonly seen in high-proliferative sarcomas, as were peritumoral edema and contrast enhancement [39].

After initial grading, radiomics can be utilized to also aid in prediction of metastasis, recurrence, disease progression, and overall prognostication. Radiomics-based machine learning methods can be utilized to predict distant metastasis for soft tissue sarcoma with greater than 90% accuracy [40]. Distant metastasis occurs in roughly 30% of soft tissue sarcoma patients and is a poor prognostic factor, making these predictions crucial in the determination of an appropriate initial treatment strategy [40].

Hu et al. demonstrated the ability of MRI radiomic features to retrospectively predict lung metastases from soft tissue sarcomas, with a nomogram incorporating radiomic and clinical data achieving an AUC of 0.894 and outperforming evaluation of tumor margins alone (AUC 0.666) [41]. Other studies have found that IQR and kurtosis analysis of radiomics features can also predict prognosis and recurrence, with AUCs of 0.78–0.79, albeit with lower accuracy (86%) [42]. Finally, in multi-institutional cohorts, radiomics models utilizing initial baseline MRI data have been able to identify postoperative progression (recurrence of resected tumors) of soft tissue sarcomas [43].

Taken together, these studies highlight the emerging importance of radiomics analysis in assessing tumor potential for aggressive biologic behavior, often outperforming clinical and histologic predictors when compared head-to-head. Long-term studies will be needed to support these early investigations and validate radiomics as a robust predictor of long-term disease-specific survival.

## 4. Radiomics in Treatment Planning and Assessment of Treatment Response

For radiomics to play a larger role in the influence of musculoskeletal radiology in orthopedic oncologic decision-making, there must be clear empiric benefit of radiomics-based analyses in guiding treatment decisions about surgery, chemotherapy, or radiation.

Interestingly, certain radiomics models may be valuable for specifically predicting whether tumors harbor tertiary lymphoid structures, which is difficult to ascertain with standard needle biopsy [44]. As a “digital biopsy,” radiomics could thus help identify those sarcomas that might respond to immunotherapy [45]. This could radically alter the treatment paradigm of using frontline cytotoxic chemotherapy (often doxorubicin and ifosfamide) in the neoadjuvant setting for many soft tissue sarcomas.

Radiomics analysis has shown clear benefit in the evaluation of the response of soft tissue sarcomas to treatment and can demonstrate a broad range of possible treatment responses (Figure 3, Figure 4, Figure 5 and Figure 6). For patients undergoing neoadjuvant chemotherapy before resection of a high-grade soft tissue sarcoma, a decrease in peritumoral edema and several other shape and texture features were associated with response to therapy [46]. Other studies have been able to extract radiomic features from pre- and post-neoadjuvant chemotherapy MRIs of soft tissue sarcomas and analyze the delta-radiomics (changes in features) using machine learning methods to separate responders to therapy from non-responders [47]. This offers great utility in predicting treatment response early in the course of therapy. MRI volumetric and texture analysis can also be utilized to predict tumor behavior and response to therapy in locally aggressive tumors such as desmoid fibromatosis [48]. While preliminary investigations into standard deviation measurements on ADC maps were found to be a crude biomarker of tumor heterogeneity [49], Valenzuela et al. were able to expand on this use of ADC maps and predict a pathologic treatment response utilizing radiomics from quantitative diffusion-weighted MRI and ADC maps for undifferentiated pleomorphic sarcomas of the extremity [50]. Because ADC maps produce absolute measurements of tissue diffusivity, the maps’ absolute values are comparable across different patients and scanners, making the use of ADC-based radiomic features a promising approach to guide early treatment decisions.

## 5. Current Challenges

The reproducibility and validation of radiomics studies in soft tissue sarcomas remain a critical issue. Recent systematic reviews have found that only 59% of studies assessed feature reproducibility, 69% conducted internal validation, and less than 25% included external validation [51]. This gap in model generalizability highlights the need for standardized multi-center imaging and radiomics protocols, and to a greater extent, international guidelines to guide radiomics analysis for soft tissue sarcomas. Similar to the guidelines proposed for standardization of radiomic features proposed by the Image Biomarker Standardization Initiative, further guidelines must be established for imaging protocols and radiomics workflows [52].

Soft tissue sarcoma patients often undergo follow-up imaging across different scanners and institutions, with variations in acquisition parameters (slice thickness, contrast phase, scanner model, etc.). These variations introduce technical variability that can confound radiomic feature extraction and reduce model effectiveness (Table 1) [53]. For a smoother analysis to occur, two technical adjustments must occur. First is the need for image resampling, which adjusts the spatial resolution and voxel size of medical images to harmonize them for analysis or combine different datasets [54]. Second is data resampling, which addresses class imbalance (uneven distribution of various tumor types) in the extracted radiomic features by oversampling or undersampling the patient cohorts to improve machine learning model performance [55].

Though radiomics routinely extracts quantitative features from single sequences, many modern studies are incorporating multi-sequence MRI data to better capture tumor heterogeneity. However, there are different ways in which multi-sequence MRI data are integrated into radiomics models. Studies can utilize a fusion model to concatenate features from each sequence into a feature pool and treat them all equally [56], use a model ensemble or weighted output strategy where separate models are used for each sequence and then combined via weighted summation [57], or use an attention-based channel weighting system which fuses multi-sequence radiomics features into a single representation [58]. Similarly, differing methods exist on how to weight or normalize features, including sequence-specific normalization, inter-sequence weighting during fusion, and inter-observer stability filtering (to exclude certain features) [41,56]. This variability in how features from multi-sequence MRI images are integrated, weighted, and normalized must be addressed in order to develop study reproducibility and generalizability across mutliple centers.

Another challenge that may slow the development of radiomics-based detection methods for many soft tissue sarcomas is that many of the subtypes are particularly rare, with limited data and small samples available, even at large sarcoma centers. With the more common sarcoma subtypes arises the issue of the sheer amount of data to be processed for multiple imaging studies and many patients. Statistical strategies must also be developed to mitigate the chance of false discovery, and to prevent high rates of false positives in tumor diagnosis algorithms.

Related to these dataset limitations are other issues of feature dimensionality and interpretability. Radiomics analysis in soft tissue sarcoma typically yields thousands of quantitative features from multi-sequence MRI, often exceeding the sample size and risking overfitting while reducing model interpretability and generalizability. Studies can address this by applying filtering for reproducibility and redundancy such as intraclass correlation coefficient (ICC) thresholds and pairwise correlation analysis, as well as feature-selection or dimensionality-reduction methods such as least absolute shrinkage and selection operator (LASSO), minimum redundancy–maximum relevance (mRMR), or principal component analysis (PCA) [41,56,59].

**Table 1 cancers-17-03661-t001:** Examples of radiomic features of varying types with demonstrated significance in soft tissue sarcoma. Abbreviations: GLCM (gray-level co-occurrence matrix).

Radiomic Feature (Type)	Definition of Feature	Significance to Sarcoma Imaging/Notes on Robustness
Tumor size/volume (shape)	3D volume of segmented region of interest, representing whole tumor	Predictor for soft tissue sarcoma recurrence and metastasis, but influenced by segmentation protocols [51,60]
Mean intensity (first-order)	Average voxel intensity (histogram) within region of interest	Used in CT/MRI sarcoma models for grade and prognosis, but affected by image acquisition techniques [61]
Skewness/Kurtosis (first-order)	Asymmetry of intensity distribution (skewness) and peakedness relative to normal (kurtosis)	Can help predict local recurrence and metastases; sensitive to many variables including outliers and binning [60]
GLCM Contrast (texture)	Intensity contrast between voxel and its neighbor, measuring local heterogeneity	Can predict tumor heterogeneity and thus histologic grade and aggressiveness, but sensitive to many variables [62,63]
Peritumoral textural summary (region-based)	Summary of features capturing characteristics of peritumoral zone	Can describe peritumoral microenvironment and improve models that predict grade of sarcomas; sensitive to segmentation variability [37]
GLCM Entropy/Dissimilarity (texture)	Texture entropy/dissimilarity—measures of heterogeneity	Higher values are related to high grade and more aggressive sarcomas; these features can vary with binning [63,64,65]

Further challenges exist with the labor-intensive process of tumor segmentation, and difficulty of integrating radiomics into the clinical workflow available on most commercial PACS. Unlike 3D reconstructions in CT/MRI, no reimbursement or RVU (relative value unit) credit is available for radiomics analysis, discouraging adoption into radiologists’ routine workflows outside of oncologic imaging research.

As a whole, while the reproducibility and validation of radiomics studies are improving, for radiomics to be implemented into the daily workflow in a widespread manner, more multicenter prospective radiomics studies that follow standardized protocols are required to truly establish radiomics as a biomarker. Moreover, administrative and logistical challenges of incorporating radiomics into or alongside PACS systems must be overcome to align physician and hospital incentives toward improving patient outcomes.

## 6. Discussion and Future Perspectives

The future of radiomics in imaging and management of soft tissue sarcoma involves integration into the daily workflow in the reading room. If a patient’s current imaging study is able to be segmented in real time, with automation of the segmentation process and comparison of the results to the patient’s prior imaging studies, a radiologist could include in their report information about potential tumor growth or treatment response. Alternatively, for a patient presenting with a new bone tumor or soft tissue mass, insights integrated into the PACS may offer suggestions of differential diagnoses and likelihood of malignancy that the radiologist can incorporate into their report.

Machine learning and AI can enhance multiple aspects of radiomic analysis, including analyzing large amounts of radiomic data in a shorter time, automating the segmentation and radiomic feature extraction process, and even offering data-driven clinical insights the radiologist can incorporate into their reports. Segmentation automation has benefited from the introduction of foundation models such as the Segment Anything Model (SAM) and its medical derivatives (MedSAM), which offer region-of-interest delineation across modalities and institutions and can potentially reduce interobserver variability and manual labor [66]. Hybrid radiomics—deep learning approaches, which combine radiomic features with high-level deep representations learned by neural networks—also carry the potential to improve predictive performance and generalizability across cancer types [67,68]. Machine learning and AI can also assist in both habitat imaging, the extraction of subregional features that reflect intratumoral heterogeneity (to better correlate with necrosis, cellularity, and vascularity), as well as delta-radiomics, the measure of temporal changes in radiomic features across serial imaging to assess early treatment response before gross morphologic changes may be visible [36,69].

As radiomics transitions to validation-driven research, reproducibility, external validation, and potential of integration into PACS should be the focus of future studies. Prospective studies should evaluate the ability of radiomics’ protocols to yield insights into diagnosis and management of soft tissue sarcomas at a multicenter level, and explore associations with other “-omics” data such as genomics and metabolomics. Such studies should highlight how radiomics can enhance the complex multidisciplinary care of soft tissue sarcoma patients to ultimately improve outcomes. Finally, as radiomics approaches translation into clinical practice, health economic evaluation will become essential to justify adoption. Cost-effectiveness and workflow efficiency studies should assess how radiomics integration can reduce diagnostic uncertainty, prevent unnecessary biopsies, or enable earlier treatment initiation, all to improve outcomes while minimizing cost.

## 7. Conclusions

Radiomics represents a vital step toward precision imaging in soft tissue sarcomas, offering quantitative and reproducible biomarkers that complement traditional interpretation of CT and MRI imaging. By extracting complex quantitative information from standard imaging, radiomics can help with tasks that often challenge conventional qualitative imaging assessment, such as enhancing diagnostic accuracy, aiding in tumor grading, and predicting treatment response, thus informing prognostication. However, widespread clinical adoption is limited by issues regarding standardization, reproducibility, and integration into clinical workflows. These can be addressed through standardized imaging protocols and model validation methods, best achieved through cross-institution collaboration. As advances in machine learning and artificial intelligence continue to aid in automation and interpretation, radiomics will grow from a research tool into a clinical resource that can augment every aspect of soft tissue sarcoma management with the goal of improving patient outcomes.

## Figures and Tables

**Figure 1 cancers-17-03661-f001:**
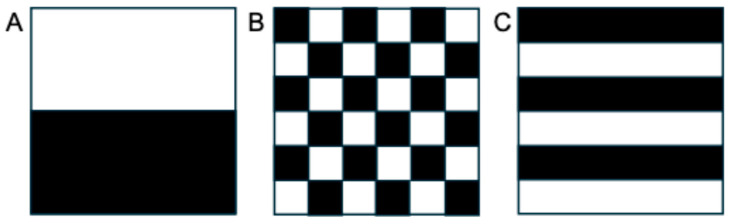
Although appearing quite different with varying textural features, these images with a (**A**) block, (**B**) checkerboard, and (**C**) striped pattern would have the same overall mean and standard deviation when analyzed globally, because they are composed of the same number of white and black pixels. These principles apply to global analysis of tumors as well, highlighting the need for analysis of other features that measure spatial heterogeneity and can be extracted from routine clinical imaging data.

**Figure 2 cancers-17-03661-f002:**
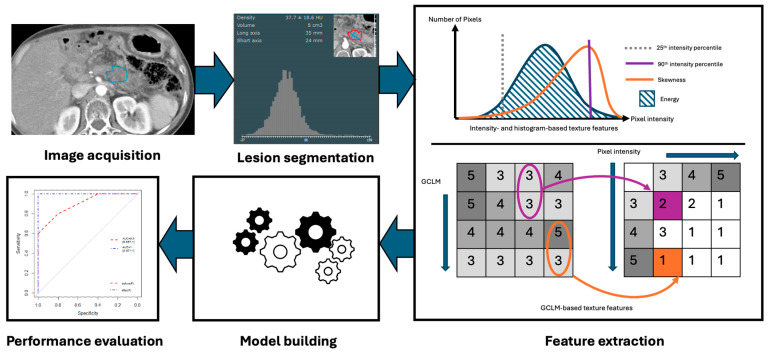
Radiomics workflow, including imaging acquisition and segmentation of lesions, followed by feature extraction and development of models.

**Figure 3 cancers-17-03661-f003:**
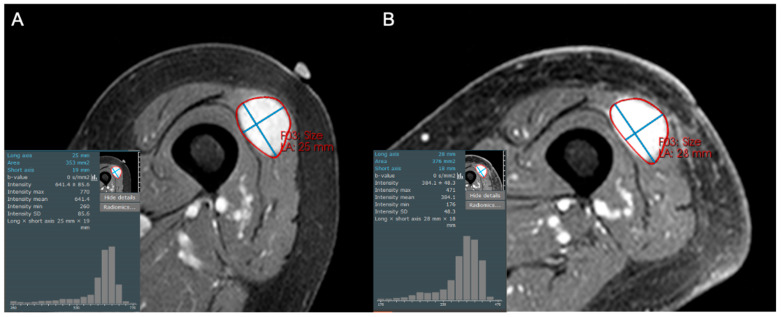
Axial contrast-enhanced fat-suppressed T1-weighted images of an 84-year-old female with a high grade leiomyosarcoma (grade 3/3), both (**A**) before and (**B**) after 50 Gy neoadjuvant radiotherapy. The post-treatment MRI reveals persistent solid enhancement, with no discernable shift in pixel intensities or decreased central enhancement to suggest internal necrosis. This was confirmed at resection, with histologic necrosis estimated to be only 5%.

**Figure 4 cancers-17-03661-f004:**
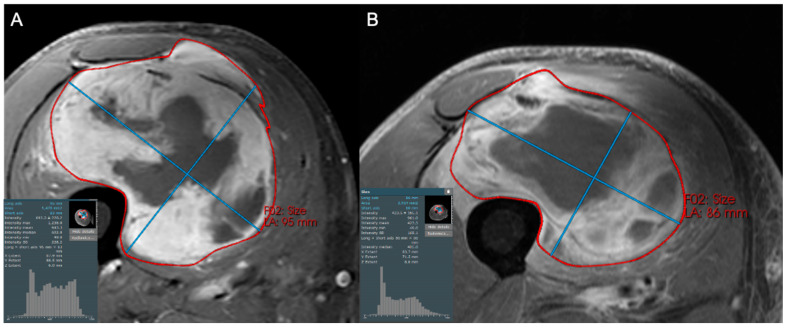
Axial contrast-enhanced fat-suppressed T1-weighted images with associated histogram of pixels of a 69-year-old male diagnosed with pleomorphic rhabdomyosarcoma, high grade (3/3). (**A**) Initial and (**B**) follow-up MRI after neoadjuvant therapy (chemotherapy and radiation) showing minimal change in size but diminished enhancement and thickness of the peripheral rind of the tumor. Such changes are traditionally described only qualitatively, but can be quantified with first order image texture statistics such as skewness, manifested as a leftward shift in the pixel intensity histogram. Higher order image texture features further describe the spatial arrangement and heterogeneity of pixel values, allowing for more nuanced tumor response assessment.

**Figure 5 cancers-17-03661-f005:**
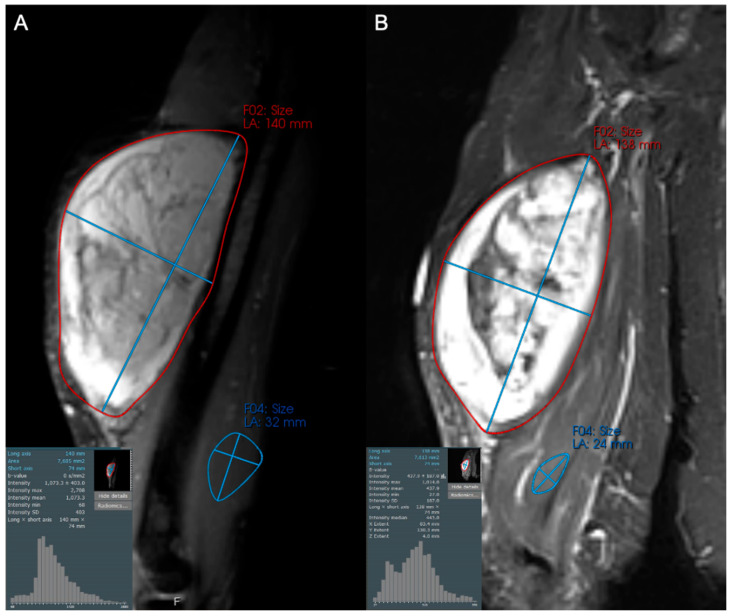
Coronal STIR images of a 57-year-old female with a large arm mass, diagnosed as dedifferentiated liposarcoma, high grade (3/3) on biopsy. Compared to baseline (**A**), the post-treatment examination (**B**) shows little change in maximum tumor diameter but a subtle increase in hypointense internal components (increase in the left tail of the histogram). This change likely reflects post-treatment tumor hyalinization and internal hemorrhage, but is difficult to capture and quantify with traditional response assessment instruments like RECIST1.1 and Choi criteria. At resection, only 20% of the tumor was viable.

**Figure 6 cancers-17-03661-f006:**
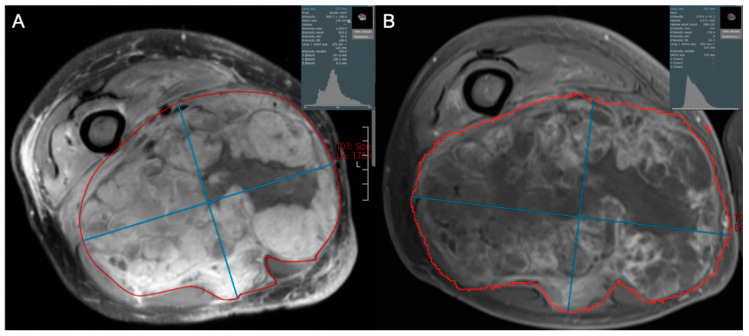
Axial contrast-enhanced fat-suppressed T1-weighted images of a 62-year-old male diagnosed with high grade pleomorphic spindle cell (fibroblastic) sarcoma. (**A**) Initial and (**B**) post-treatment images demonstrate that 2 cycles of doxorubicin and dacarbazine with radiotherapy resulted in a small decrease in tumor size, but a marked decrease in tumor enhancement. This is reflected by increased skewness and leftward shift in pixel intensities, and suggests an excellent response to therapy, ultimately confirmed as 98% tumor necrosis at resection.

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
