# Peer review of "Radiomics in Soft Tissue Sarcoma: Toward Precision Imaging in Oncology"

_cancers, 2025, doi:10.3390/cancers17223661_

Round 1

Reviewer 1 Report

Comments and Suggestions for Authors

Thankyou for the opportunity to review this overview of the current state of play of radiomics in soft tissue sarcoma.

I found the article generally well written and informative.

I have a few comments.

Line 149 mentions a "large number of studies" that have established unique radiomic features separating leiomyosarcoma and leiomyomas but then references only 1 study - that of Lakhman. Are there other studies that could be  referenced here usefully and why was this particular study chosen?

Line 209: there is an extra "to" that should be removed-  ie should read "to be applied across...."

Lines 245-247: could the "clear benefit" include reducing radiologists time/reducing reporting costs as well as improved treatment decisions?

Line 254/5: something is missing in the first sentence of this paragraph.  Please check. Should it read "Radiomics has shown clear benefit in the evaluation of response of soft tissue sarcomas to treatment..."?

Line 265: "Advancing over preliminary investigations.... "please check wording. It doesn't quite make sense at the moment. 

Discussion: The authors helpfully describe some of the work needed to help shift radiomics into routine work flows. Are there any specific examples of use of radiomics at multicentre level that are already in use that the authors could point to?

Should forms of health economic evaluation be included as part of research into the translation of radiomics techniques into routine clinical care? This is not mentioned in the article at present.

Author Response

Comment 1: Line 149 mentions a "large number of studies" that have established unique radiomic features separating leiomyosarcoma and leiomyomas but then references only 1 study - that of Lakhman. Are there other studies that could be  referenced here usefully and why was this particular study chosen?

Response 1: Thank you for your comment. We used the Lakhman study simply as a representative example, and changed the sentence to indicate it is only one example. Many of the other studies are referenced later in the paragraph and also demonstrate similar findings.

Comment 2: Line 209: there is an extra "to" that should be removed-  ie should read "to be applied across...."

Response 2: Thank you for this comment, we have made this correction (line 228).

Comment 3: Lines 245-247: could the "clear benefit" include reducing radiologists time/reducing reporting costs as well as improved treatment decisions?

Response 3:  Thank you for this comment. We agree that reducing radiologists’ time and reporting costs as well as lower reporting costs are all benefits of using radiomics to improve decision making. We realize there may have been a miscommunication in this phrase – what was meant to be said was that for radiomics to influence orthopedic oncologic decision-making, there needs to be a clear benefit in the data/literature that shows it improves decision making on treatments. We have adjusted the sentence to indicate that this is was we meant.

Comment 4: Line 254/5: something is missing in the first sentence of this paragraph.  Please check. Should it read "Radiomics has shown clear benefit in the evaluation of response of soft tissue sarcomas to treatment..."?

Response 4: Thank you for this comment. We have made this correction.

Comment 5: Line 265: "Advancing over preliminary investigations.... "please check wording. It doesn't quite make sense at the moment. 

Response 5: Thank you for this comment. We clarified the first half of this sentence to refine its meaning.

Comment 6/7: Discussion: The authors helpfully describe some of the work needed to help shift radiomics into routine work flows. Are there any specific examples of use of radiomics at multicentre level that are already in use that the authors could point to?

Should forms of health economic evaluation be included as part of research into the translation of radiomics techniques into routine clinical care? This is not mentioned in the article at present.

Response 6/7:  Thank you for these recommendations regarding our discussion. Although we were not able to find specific examples of the use of radiomics at the multicenter level, we had considered including sections on the Cancer Imaging Archive (TCIA) and EUCAIM cancer imaging infrastructure but ultimately did not include them as they had not yet integrated radiomics into their platforms. However, we did add a section to the discussion on health economic evaluation, and agree that this should be included in research on the translation of radiomics into clinical care.

Reviewer 2 Report

Comments and Suggestions for Authors

 This manuscript provides a clear and comprehensive overview of the application of radiomics in soft tissue sarcoma (STS). It effectively summarizes the motivation, current evidence, and clinical potential of radiomics for diagnostic, prognostic, and therapeutic purposes. The writing is overall clear and logically structured, and the figures and references are relevant.
 However, the manuscript would benefit from a more critical appraisal of existing limitations in radiomics research, clarification of several technical terms, and broader introductions beyond certain narrowly focused examples. 

Abstract
- The example of differentiating uterine leiomyosarcoma from leiomyoma is too narrow for an abstract summary. 
- The phrase “broader adoption” is vague. Specify whether this refers to validation across larger, multi-institutional datasets, or clinical integration.
- The term “novel quantitative imaging biomarkers” is ambiguous. If these indicate "Diffusion-weighted or contrast-enhanced MRI" are now well-established techniques and not “novel” anymore. 

Introduction
- The introduction is overly dense. The current abstract allocates disproportionate emphasis to parametric imaging (e.g., DWI, DCE), which detracts from the main topic of radiomics.
- Include CT as part of the diagnostic pathway, since CT-based radiomics is discussed later in the manuscript
- There are limited citations supporting general radiomics concepts. Include foundational references.
- The sentence conflates pixel intensity and signal intensity. Correct wording should read:
“...signal intensity (in MRI) or pixel intensity quantified as Hounsfield units (in CT)...”
- “tumor heterogeneity” should not encompass size or margin definition. These are morphological, not heterogeneity, descriptors. 

Radiomics in Initial Diagnosis of Soft Tissue Sarcomas
- The section focuses almost exclusively on uterine leiomyosarcoma vs leiomyoma. Consider incorporating additional diagnostic usages—such as desmoid-type fibromatosis vs sarcoma, benign peripheral nerve sheath tumors, or myxoid vs pleomorphic sarcomas.
- The statement that “radiomics models were comparable to experienced radiologists” may overstate performance given the small sample size (78 patients). Temper this by noting that such findings are preliminary and derived from limited, single-center cohorts.

Current Challenges
- Expand the discussion on how multi-sequence MRI data are integrated in radiomic models. Clarify whether features from different sequences are handled in an ensemble or fused structure, and how feature weighting or normalization is managed.
- Many extracted radiomic features are redundant or correlated. Suggest elaboration on feature selection and dimensionality reduction techniques to improve interpretability and prevent overfitting.
- Address the imbalance between feature set and dataset sizes in STS research, emphasizing the importance of multicenter data sharing and harmonization.

Discussion
- Consider changing subheading into "Discussion and Future Perspectives"
- The discussion should be expanded to reflect recent advances that improve the efficiency and clinical feasibility of radiomics
 1) Segmentation automation: Integration of foundation models such as Segment Anything Model (SAM) for robust ROI definition.
 2) Hybrid approaches: Combination of handcrafted radiomic features with deep learning representations for improved performance and generalizability.
 3) Habitat imaging - Subregional feature analysis capturing intra-tumoral heterogeneity.
 4) Delta-radiomics
 5) A forward-looking section could emphasize radiogenomic integration and alignment of radiomics with AI-driven precision oncology strategies.
- The conclusion should more explicitly discuss clinical translation barriers (regulatory validation, interpretability, standardization) and not only potential benefits.

Author Response

Comment 1: Abstract
- The example of differentiating uterine leiomyosarcoma from leiomyoma is too narrow for an abstract summary. 
- The phrase “broader adoption” is vague. Specify whether this refers to validation across larger, multi-institutional datasets, or clinical integration.
- The term “novel quantitative imaging biomarkers” is ambiguous. If these indicate "Diffusion-weighted or contrast-enhanced MRI" are now well-established techniques and not “novel” anymore.

Response 1:  Thank you for these recommendations. We adjusted the abstract to remove the specific leiomyosarcoma/leiomyoma example, clarify the phrase “broader adoption”, and clarified that “novel quantitative imaging biomarkers” refer to radiomics features that may be predictors for diagnosis, grading, etc.

Comment 2: Introduction
- The introduction is overly dense. The current abstract allocates disproportionate emphasis to parametric imaging (e.g., DWI, DCE), which detracts from the main topic of radiomics.

Response 2: Thank you for this recommendation. We shortened the sections you referenced to only briefly mention parametric imaging, so that radiomics remains the main focus of the introduction.

Comment 3: - Include CT as part of the diagnostic pathway, since CT-based radiomics is discussed later in the manuscript

Response 3: Thank you for this comment, we have added CT into the diagnostic pathway.

Comment 4: - There are limited citations supporting general radiomics concepts. Include foundational references.

Response 4: Thank you for your comment. We have added more citations explaining basic radiomics concepts, including those by Lambin, Aerts, and Gillies.

Comment 5: - The sentence conflates pixel intensity and signal intensity. Correct wording should read:
“...signal intensity (in MRI) or pixel intensity quantified as Hounsfield units (in CT)...”

Response 5: Thank you for this comment. We edited this sentence to clarify the difference between these two terms.

Comment 6: - “tumor heterogeneity” should not encompass size or margin definition. These are morphological, not heterogeneity, descriptors.

Response 6: Thank you for this suggestion. We agree, and have modified this section to indicate that size and margin refer to tumor morphology rather than heterogeneity.

Comment 7: Radiomics in Initial Diagnosis of Soft Tissue Sarcomas
- The section focuses almost exclusively on uterine leiomyosarcoma vs leiomyoma. Consider incorporating additional diagnostic usages—such as desmoid-type fibromatosis vs sarcoma, benign peripheral nerve sheath tumors, or myxoid vs pleomorphic sarcomas.

Response 7: Thank you for this comment. We added additional diagnostic use cases of radiomics to this section, including references for desmoid-type fibromatosis, benign vs malignant PNSTs, and benign myxomas vs malignant myxoid sarcomas.

Comment 8: - The statement that “radiomics models were comparable to experienced radiologists” may overstate performance given the small sample size (78 patients). Temper this by noting that such findings are preliminary and derived from limited, single-center cohorts.

Response 8: Thank you for this suggestion. We have edited the sentence to clarify these findings were drawn from smaller samples /single-center cohorts.

Comment 9: Current Challenges
- Expand the discussion on how multi-sequence MRI data are integrated in radiomic models. Clarify whether features from different sequences are handled in an ensemble or fused structure, and how feature weighting or normalization is managed.

Response 9: Thank you for this suggestion. We added clarification as to how multi-sequence MRI data can be integrated into radiomics models, including discussion on fusion and feature weighting/normalization.

Comment 10/11: - Many extracted radiomic features are redundant or correlated. Suggest elaboration on feature selection and dimensionality reduction techniques to improve interpretability and prevent overfitting.

- Address the imbalance between feature set and dataset sizes in STS research, emphasizing the importance of multicenter data sharing and harmonization.

Response 10/11:  Thank you for these suggestions. We added portions to the section on feature selection / dimensionality reduction techniques as well as the issue of discordance between feature set size and sample size.

Comment 12: Discussion
- Consider changing subheading into "Discussion and Future Perspectives"
- The discussion should be expanded to reflect recent advances that improve the efficiency and clinical feasibility of radiomics
 1) Segmentation automation: Integration of foundation models such as Segment Anything Model (SAM) for robust ROI definition.
 2) Hybrid approaches: Combination of handcrafted radiomic features with deep learning representations for improved performance and generalizability.
 3) Habitat imaging - Subregional feature analysis capturing intra-tumoral heterogeneity.
 4) Delta-radiomics
 5) A forward-looking section could emphasize radiogenomic integration and alignment of radiomics with AI-driven precision oncology strategies.

Response 12: Thank you for this comment. We updated the subheading to “discussion and future perspectives”. We also included some more recent advances including the SAM and segmentation automation portion, hybrid approaches, habitat imaging, and delta-radiomics, and discussed the ways in which AI can improve these processes.

 Comment 12: - The conclusion should more explicitly discuss clinical translation barriers (regulatory validation, interpretability, standardization) and not only potential benefits.

Response 12: Thank you for this comment. We made sure to include in our discussion and conclusion a section about the barriers to clinical translation.

Reviewer 3 Report

Comments and Suggestions for Authors

This paper is very interesting, but I believe it is very important to establish a proper correlation with the pathological findings of the surgical specimen. It would be interesting if the authors presented cases with radiopathological correlation and included pathologists in this study which, although focused on radiology, it is well known that the cornerstone of diagnosis remains the anatomopathological evaluation.

Author Response

Comment 1: This paper is very interesting, but I believe it is very important to establish a proper correlation with the pathological findings of the surgical specimen. It would be interesting if the authors presented cases with radiopathological correlation and included pathologists in this study which, although focused on radiology, it is well known that the cornerstone of diagnosis remains the anatomopathological evaluation.

Response 1:  Thank you for your comment. We agree that it is important to include cases with rad-path correlation. As such, we have included a pathologist, Dr. Ortiz Requena, and several cases that highlight correlations with pathological findings.  Specifically, figures 3, 4, 5, and 6 are all meant to show imaging with a description of either the pathology from biopsy or pathology from surgical resection, and the correlation between pathologic diagnosis and imaging appearance / radiomic data. We have ensured that each figure caption highlights the pathologic diagnosis for each case.

Round 2

Reviewer 2 Report

Comments and Suggestions for Authors

All reviewer comments have been carefully addressed, and the previously omitted parts have been sufficiently discussed. The authors’ thorough revisions have  enhanced the quality, clarity, and coherence of this review paper. Thank you.

Reviewer 3 Report

Comments and Suggestions for Authors

Thank you for the response